# Effects of intraoperative neuromonitoring (IONM) technology on early recovery quality in patients after thyroid surgery: A randomized controlled trial

Haocong Chen [1], Zhijun Lu [2]*

1 Department of Anesthesiology, Tongren Hospital, Shanghai Jiao Tong University School of Medicine, Shanghai, China, 2 Department of Anesthesiology, Ruijin Hospital Affiliated to Shanghai Jiao Tong University School of Medicine, Shanghai, China

* lusamacn@126.com

## Abstract

### Background

Patient-focused evaluation of postoperative recover has been recognized as one of the most important concerns in postoperative medicine. Previous studies have shown that the Quality of Recovery-40 (QoR-40) Questionnaire can be used to accurately assess the quality of recovery from surgery. During thyroid surgery using intraoperative neuromonitoring (IONM) technology, the strategy of low dose of muscle relaxant, intubation of different endotracheal tubes and electrical stimulation on vocal cord are applied. Its still unknown if these performances would affect patients' postoperative recovery in thyroid surgery patients.

### Methods

82 patients were randomly assigned to the neuromonitoring group (NEURO Group) and the control group (CON Group). In the CON Group, rocuronium (0.6 mg / kg) was given for intubation and additional dose was injected if needed, while in the NEURO Group, only rocuronium (0.3 mg / kg) was given when induction. The primary outcome is the QoR-40 scores on postoperative day 1 (POD1) and postoperative day 3 (POD3). Other parameters, such as postoperative nausea or vomiting (PONV) and medical cost were also recorded.

### Results

One subject in each group was excluded, leaving 80 for analysis. In the NEURO Group, the global QoR-40 score, emotional state, physical comfort, physical independence and pain were significantly lower both on POD1 and POD3 ($P<0.05$). Patients in the NEURO Group had a higher incidence of PONV ($P<0.05$) and medical expense ($P<0.05$).

**Data Availability Statement:** All relevant data are within the paper and its Supporting information files.

**Funding:** The author(s) received no specific funding for this work.

**Competing interests:** The authors have declared that no competing interests exist.

## Conclusions

After thyroidectomy, the patients using IONM suffer worse quality of recovery, more risk of PONV and increased medical expense.

## Introduction

Postoperative recovery is a key concern for patients undergoing surgery. Delayed postoperative recovery may cause a patient discomfort, a longer hospital stay, a delayed return-to-work, and increased health care costs. Recent advances in anesthetic and surgical care mean that the quantitative analysis of fragmentary indicators of postoperative recovery, including morbidity or mortality rates, cannot accurately reflect patients' recovery profiles. Rather, overall assessments of quality of life of patients who are undergoing surgery provide more appropriate measurements that facilitate investigations into the effects of anesthetic or surgical care on patients' recovery and their satisfaction. The Quality of Recovery-40 (QoR-40) Questionnaire (S1 Table.) is a reliable multidimensional instrument to evaluate patients' health status after surgery and anesthesia [1, 2]. Previous studies have shown that the QoR-40 can be used to accurately assess the quality of recovery from surgery [3, 4].

With the increasing pace of the modern life, the incidence of thyroid diseases in China has been increasing year by year [5]. According to the latest tally show, Chinese patients suffering from thyroid diseases increased significantly up to 36.9% [6]. One of the most severe complications of thyroid surgery is recurrent laryngeal nerve palsy (RLNP). In cases of bilateral cord involvement, RLNP can cause vocal fold paralysis as well as dysphonia, difficulty swallowing, and respiratory problems including aspiration symptoms and airway obstruction [7].

With the increasing use of artificial intelligence in anesthesia equipment, many use of monitoring with the bi-spectral index has shown benefits reducing time to extubation, orientation in time and place, and discharge from both the operating room [8–10]. Intraoperative neuromonitoring (IONM) technology was proposed as a means of verifying the functional integrity of the RLNP and it can rapidly locate the laryngeal nerve and protect the functional integrity of it, thereby minimizing the risk of recurrent laryngeal nerve palsy. The waveform, amplitude, threshold and time latency of electromyography (EMG) were analyzed to determine the integrity of nerve function [11]. Nevertheless, compared to traditional thyroid surgery, there are some differences during surgery with IONM. One of these is that a smaller dose of muscle relaxant should be used when IONM applied because the EMG amplitude is sensitive to neuromuscular blockade. Besides, the tracheal tube with larger outer diameter is used for intubation and the patient's vocal cord is intermittently stimulated during the procedure with IONM. It is still unknown whether these changes will affect the early postoperative recovery quality in thyroid surgery patients.

This randomized controlled study aimed to evaluate the effects of IONM technology on early postoperative recovery quality by conducting the QoR-40 Questionnaire in thyroid surgery patients. We hypothesized that IONM technology would reduce the early recovery quality of thyroid surgery patients.

## Materials and methods

### Design and patients

This single-center, prospective randomized controlled research was conducted at Ruijin Hospital Affiliated to Shanghai Jiao Tong University School of Medicine from March 1st 2021 to September 30th 2021. Our research was approved by the Ethics Committee of Ruijin Hospital

Affiliated to Shanghai Jiao Tong University School of Medicine [Ethics Committee Reference Number: (2020) Ethical Review Scientific Research No.173] and our research was prospectively registered in the Chinese Clinical Trial Registry on February 27, 2021 (registration number: ChiCTR2100043738). Each patient signed an informed consent form before to surgery. The Consolidated Standards of Reporting Trials (CONSORT) recommendations (CONSORT Checklist) are followed in our study; a CONSORT flow diagram is shown in Fig 1.

The gender of the patients had no bearing on their selection, which was limited to patients aged 20–70 years with a BMI of 18–30 kg/m2 and American Society of Anesthesiologists (ASA) grades of I-II. These were the exclusion criteria: ASA physical status III or more; predicted difficult intubation; previous known vocal cord paralysis; significant heart, kidney or liver disease; psychiatric or neurological disorders; allergy to any anaesthetic drug and/or written informed consent denied.

Using a web-based response system (www.random.org) and a random sequence generator process, each eligible participant was randomly assigned to either the NEURO Group or the CON Group with a 1:1 allocation.

## Anesthetic procedure

Before their surgeries, all of the participants regularly fasted and did not take any premeds. A variety of routine monitoring procedures were used, including electrocardiography (ECG), noninvasive blood pressure (NBP), peripheral pulse oximetry (SpO2), airway pressure monitoring, capnography, Narcotrend (MT MonitorTechnik GnbH & Co.KG, Narcotrend-Compact) and muscle relaxant monitor (GE Healthcare Finland Oy, SJC15300045HA). The heart rate (HR), systolic pressure (SBP), diastolic pressure (DBP) and mean arterial pressure (MAP)

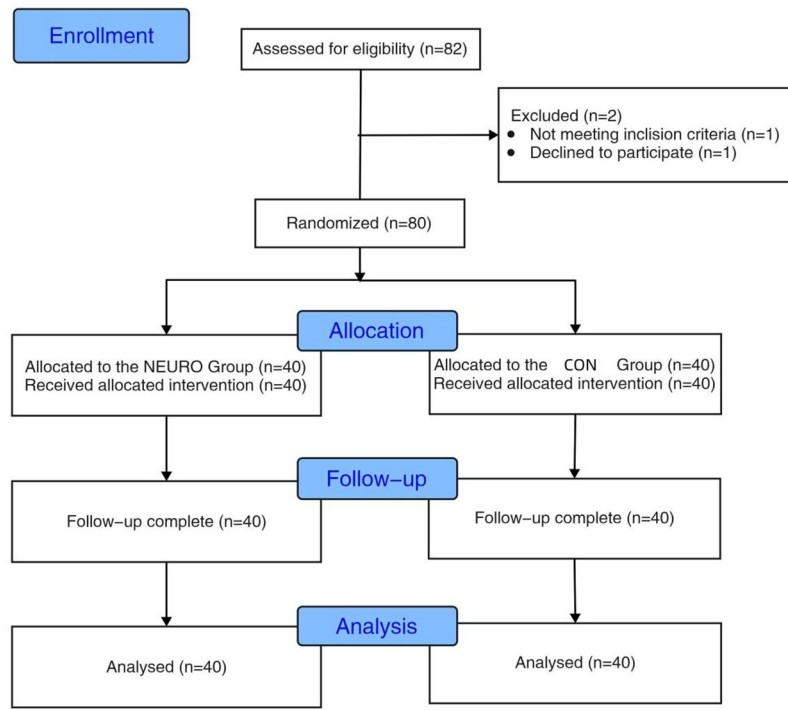

**Fig 1. CONSORT flow diagram.**

measured 10 minutes after the patients entered the room were taken as the base values. In both group, general anesthesia was induced using 0.04mg/kg midazolam, 1.5–2.5mg/kg propofol, 0.4ug/kg sufentanil and different dose of rocuronium.

Patients randomized to the CON Group received rocuronium 0.6mg/kg and traditional endotracheal tube. However, those in the NEURO group had rocuronium 0.3mg/kg, after that, The attending anesthesiologist used a video-laryngoscope to position the Never Integrity Monitor (NIM) endotracheal tube (Medtronic, USA) on the patient. The correct position of the NIM endotracheal tube was confirmed using both capnography and the IONM's dedicated signal quality checking system.

Intravenous compound general anesthesia was used to treat the patients in both groups. During surgery, anesthesia was maintained with desflurane of a Minimal Alveolar Concentration (MAC) of 0.7–0.9 and with propofol through Target Controlled Infusion (TCI) at 1.0μg/ml. Level of sedation was monitored with Narcotrend and sedatives targeted to a Narcotrend value around 37–64. Intro-operative analgesia was guaranteed by continuous infusion of remifentanil at 0.05–0.1μg/(kg·min). No additional rocuronium was added to the patients in the NEURO Group during the operation, while in the CON Group, 1/3 induction dose of rocuronium was added when the TOF = 2.

If the mean arterial pressure (MAP) of the patient during anesthesia is less than 55 mmHg or its reduction exceeds the base value by more than 20%, intravenous injection of ephedrine 5 mg was given, and the infusion speed of remifentanil was slowed down or the concentration of desflurane was reduced as appropriate; If the mean arterial pressure (MAP) during anesthesia is higher than 110mmHg or exceeds the base value by more than 20%, 5μg sufentanil is injected intravenously; If the heart rate of the patient is less than 50 beats/minute, atropine 0.5mg shall be injected intravenously. 0.25mg palonosetron for participants to prevents postoperative nausea or vomiting (PONV) and 50mg flurbiprofen axetil was provided at the end of the surgery for postoperative analgesia.

## Outcome measures

The QoR-40 Questionnaire is a valid and reliable tool that is extensively used to assess the quality of postoperative recovery [12, 13]. 40 items totaling five dimensions—emotional state, physical comfort, psychological support, physical independence, and pain—make up the QoR-40 Questionnaire. Each item is rated on a 5-point Likert scale (1 = none of the time, 2 = some of the time, 3 = usually, 4 = most of the time, 5 = all of the time) and the overall scores range from 40 to 200 [14–16].

The primary outcome of the research was the global QoR-40 score on postoperative day 1 (POD1) and postoperative day 3 (POD3). On the day before surgery, POD1 and POD3, between 6 and 8 PM, a particular researcher who was not aware of the patient groups' assignments visited each patient to conduct the QoR-40 survey. Patients filled out the survey in person while participating in a study and it was then checked to make sure all of the questions had been addressed.

The secondary outcomes included HR, SBP, DBP and MAP of immediately after intubation (T0), 1 minute after intubation (T1), 3 minutes after intubation (T2) and 5 minutes after intubation (T3); cumulative sufentanil, remifentanil and propofol consumption; the total cost of hospitalization and the adverse events on POD1.

## Statistical analysis

The sample size was calculated using previous literature [17] and the global QoR-40 score. A clinically significant difference is defined as a change of 10 points or more on the QoR-40

Questionnaire. In our pilot study, we estimated an overall standard deviation of 13 points, and with an of 0.05, 37 patients would be needed in each group (assuming an 80% power). Since all patients were hospitalized and all studies were completed 24 hours after surgery, the dropout rate was low, estimated to be 10%. Therefore, we recruited 41 patients in each group for a total of 82 patients in this research.

For statistical analysis, SPSS version 20.0 software (SPSS Inc., Chicago, IL) was used. The mean, standard deviation, or median are used to express continuous variables (interquartile range). The t-test was used for inter group comparison if the data were normal, instead that, the non-parametric test was used to compare groups. P-values less than 0.05 were considered statistically significant.

## Results

We assessed 82 patients for eligibility to participate in this research. One patient did not meet the inclusion criteria, one declined to participate and the remaining 80 patients consented to participate. The number of patients at each level of the investigation is depicted in the flowchart in Fig 1 and the Table 1 lists the characteristics of the study population. There was no difference between the groups in terms of gender, age, height, weight, BMI, ASA physical states I/II, preoperative QoR-40 score, operation time, anesthesia time or hospitalization time.

The distribution of the QoR-40 scores and sub-scores of each dimension on POD1 and POD3 are shown in Table 2. The NEURO Group had significantly lower global QoR-40 scores and sub-scale scores for emotional state, physical comfort, physical independence, and pain than the CON Group on POD1 and POD3 (P < 0.05). The scores of psychological support did

**Table 1. Patient characteristics and surgical data.**

|  | NEURO Group | CON Group | P-Value |
|---|---|---|---|
|  | (N = 40) | (N = 40) |  |
| Gender (Male/Female) | 13 /27 | 15 /25 | 0.138 |
| Age (year) | 37.71 ± 11.41 | 40.91± 13.21 | 0.262 |
| Height (cm) | 164.11 ± 5.82 | 166.21 ± 7.69 | 0.177 |
| Weight (kg) | 63.31 ± 8.07 | 65.51 ± 11.91 | 0.123 |
| BMI (kg/m$^2$) | 24.21 ± 10.12 | 25.42 ± 10.81 | 0.634 |
| ASA physical states I/II (n) | 1/39 | 1/39 | 1 |
| Preoperative QoR-40 score |  |  |  |
| Global QoR-40 score | 182.72±1.79 | 183.63±1.36 | 0.196 |
| Emotional state | 38.40±3.20 | 38.85±2.19 | 0.555 |
| Physical comfort | 53.31±2.16 | 53.40±3.96 | 0.925 |
| Psychological support | 34.52±0.98 | 34.71±0.62 | 0.309 |
| Physical independence | 24.84±0.03 | 24.90±0.30 | 0.480 |
| Pain | 32.82±1.75 | 33.37±1.64 | 0.187 |
| Operation time (h) | 1.34 ± 0.51 | 1.30 ± 0.50 | 0.742 |
| Anesthesia time (h) | 1.84 ± 0.51 | 1.80 ± 0.62 | 0.785 |
| Hospitalization time (day) | 4.18±2.13 | 4.21 ± 2.39 | 0.941 |

Data are presented as mean ± standard or number of patients (%) where appropriate

*Abbreviation*: *ASA* American Society Anesthesiologists, *BMI* Body mass index

**Table 2. Dimensions of the QOR-40 questionnaire on POD1 and POD3.**

| | NEURO Group (N = 40) | CON Group (N = 40) | P-Value |
|---|---|---|---|
| **Global QoR-40 score** | | | |
| POD1 | 160.06 ± 4.68* | 170.42 ± 5.16 | <0.001 |
| POD3 | 170.27 ± 4.51* | 174.62 ± 4.00 | 0.006 |
| **QoR-40 Dimensions** | | | |
| **Emotional state** | | | |
| POD1 | 34.48 ± 2.61* | 36.96 ± 2.89 | 0.002 |
| POD3 | 35.46 ± 2.41* | 36.88 ± 2.10 | 0.026 |
| **Physical comfort** | | | |
| POD1 | 45.11 ± 2.67* | 52.00 ± 4.52 | <0.001 |
| POD3 | 51.37 ± 2.94* | 54.90 ± 3.60 | <0.001 |
| **Psychological support** | | | |
| POD1 | 33.55 ± 0.55 | 33.72 ± 0.94 | 0.365 |
| POD3 | 33.40 ± 0.55 | 33.48± 0.50 | 0.512 |
| **Physical independence** | | | |
| POD1 | 22.37 ± 1.37* | 24.26 ± 1.22 | <0.001 |
| POD3 | 22.71 ± 1.21* | 24.00 ± 0.87 | <0.001 |
| **Pain** | | | |
| POD1 | 26.28 ± 3.37* | 29.80 ± 3.89 | <0.001 |
| POD3 | 29.55 ± 2.93* | 31.08 ± 3.09 | 0.034 |

Notes:

* $P < 0.05$ (compared to the CON Group)

Data are presented as mean ± standard deviation

*Abbreviation*: *POD* Postoperative day, *QoR-40* Quality of recovery-40 questionnaire.

not meet statistical significance compared with either NEURO Group or CON Group on POD1 and POD3 (P > 0.05).

There was no significant difference in HR, SBP and MAP between the two groups at each time point before and after intubation, but the DBP of the patients in the NEURO Group was lower immediately after intubation (T0), 1 minutes after intubation (T1), 3 minutes after intubation (T2) and 5 minutes after intubation (T3), and the difference was statistically significant (P < 0.05; Table 3).

There was no significant difference in the highest and lowest mean arterial pressures during anesthesia (P > 0.05; Table 4).

There was no significant difference in the total dose of sufentanil and propofol in the two groups (P > 0.05). However, compared with the CON Group, the total dose of remifentanil and the total cost of hospitalization were significantly increased in the NEURO Group (P < 0.05; Table 5).

There was no significant difference between the groups in terms of throat pain (VAS>3 point), hoarseness, drinking water cough, wound pain (VAS>3 point), wound bleeding (need surgery) and fatigue (P > 0.05). On the contrary, the incidence of PONV was significantly increased in the NEURO Group (P < 0.05; Table 6).

All patients in the NEURO Group were given nerve probe to stimulate the recurrent laryngeal nerve before the end of the operation and all patients had no RLNP during the operation. All patients in the CON Group had no adverse reactions such as hoarseness and drinking water cough before leave hospital, which indirectly proved that all patients in the CON Group had no RLNP during the operation.

**Table 3. Comparison of hemodynamic changes before and after intubation.**

| | NEURO Group (N = 40) | CON Group (N = 40) | P-Value |
|---|---|---|---|
| **HR(beats/min)** | | | |
| **Basic values** | 79.91 ± 12.21 | 78.61 ± 14.21 | 0.650 |
| **Immediately after intubation (T0)** | 80.91 ± 14.82 | 81.91 ± 16.93 | 0.748 |
| **1 minute after intubation(T1)** | 71.81 ± 9.33 | 73.11 ± 16.82 | 0.687 |
| **3 minutes after intubation(T2)** | 70.92 ± 9.70 | 73.72 ± 16.22 | 0.352 |
| **5 minutes after intubation(T3)** | 69.92± 10.42 | 71.31 ± 12.02 | 0.592 |
| **SBP(mmHg)** | | | |
| **Basic values** | 129.11 ± 20.32 | 132.12 ± 21.63 | 0.473 |
| **Immediately after intubation (T0)** | 108.21 ± 22.12 | 108.13 ± 19.71 | 1 |
| **1 minute after intubation(T1)** | 96.61 ± 20.91 | 96.62± 20.92 | 0.209 |
| **3 minutes after intubation(T2)** | 96.92 ± 18.31 | 95.63± 17.53 | 0.761 |
| **5 minutes after intubation(T3)** | 93.82 ± 11.91 | 96.32 ± 11.91 | 0.351 |
| **DBP(mmHg)** | | | |
| **Basic values** | 80.11 ± 12.62 | 81.61 ± 16.22 | 0.877 |
| **Immediately after intubation (T0)** | 61.02 ± 15.61* | 69.61 ± 18.22 | 0.026 |
| **1 minute after intubation(T1)** | 54.31 ± 15.41* | 65.92 ± 14.32 | <0.001 |
| **3 minutes after intubation(T2)** | 53.41 ± 13.72* | 61.61 ± 11.82 | 0.005 |
| **5 minutes after intubation(T3)** | 59.22 ± 10.91* | 61.12 ± 13.01 | 0.001 |
| **MAP(mmHg)** | | | |
| **Basic values** | 94.71 ± 13.82 | 96.33 ± 18.11 | 0.663 |
| **Immediately after intubation (T0)** | 79.62 ± 15.92 | 77.91 ± 17.92 | 0.646 |
| **1 minute after intubation(T1)** | 71.52 ± 15.91 | 73.02 ± 15.61 | 0.666 |
| **3 minutes after intubation(T2)** | 71.12 ± 13.02 | 70.52 ± 11.71 | 0.822 |
| **5 minutes after intubation(T3)** | 70.01 ± 9.59 | 70.12 ± 11.42 | 0.975 |

Notes:

* $P<0.05$ (compared to the CON Group)

Data are presented as mean ± standard deviation

*Abbreviation*: *HR* Heart rate, *SBP* Systolic blood pressure, *DBP* Diastolic blood pressure, *MAP* Mean arterial pressure

## Discussion

We evaluated and compared the effects of IONM technology on early postoperative recovery quality by means of the QoR-40 Questionnaire in patients after thyroid surgery on POD1 and POD3. Our results demonstrate that the global score of QoR-40 Questionnaire decreased

**Table 4. Comparison of the highest and lowest MAP during anesthesia as well as the number of patients with the highest MAP exceed 20% of basic values and the lowest MAP less than 20% of basic values.**

| | NEURO Group (N = 40) | CON Group (N = 40) | P-Value |
|---|---|---|---|
| **The highest MAP(mmHg)** | 96.92 ± 14.91 | 98.20 ± 16.91 | 0.722 |
| **The lowest MAP(mmHg)** | 65.15 ± 7.60 | 66.43 ± 9.30 | 0.506 |
| **Number of patients with the highest MAP exceed 20% of basic values** | 1 (3%) | 1 (3%) | 1 |
| **Number of patients with the lowest MAP less than 20% of basic values** | 17 (43%) | 23 (58%) | 0.264 |

Data are presented as mean ± standard deviation or number (%)

*Abbreviation*: *MAP* Mean arterial pressure

**Table 5. Comparison of drug consumption in anesthesia and total hospitalization costs.**

| | NEURO Group (N = 40) | CON Group (N = 40) | *P*-Value |
|---|---|---|---|
| **Total dose of sufentanil(μg/kg)** | 0.56±0.11 | 0.52±0.11 | 0.152 |
| **Total dose of remifentanil(μg/kg)** | 9.24±4.23* | 6.68±1.98 | 0.001 |
| **Total dose of propofol(mg/kg)** | 6.36±0.40 | 6.38±0.43 | 0.840 |
| **Total cost of hospitalization(10K CNY)** | 2.62 ± 1.25* | 1.50± 0.28 | <0.001 |

Notes:

* *P*<0.05 (compared to the CON Group)

Data are presented as mean ± standard deviation

significantly in the NEURO Group, mainly in the emotional state, physical comfort, physical independence and pain perception. In addition, the incidence of the PONV, the amount of remifentanil consumption and the total cost of hospitalization was significantly increased in the NEURO Group. The difference in these aspects between group NEURO and group CON meet statistical significance difference.

At present, there are few studies on evaluating the early recovery quality of thyroid surgery patients by QoR-40 Questionnaire. The results of Myoung's research [17] show that the total score of QoR-40 after traditional thyroid surgery is 179.4, and our result is 173, which was similar to it. However, early recovery quality of patients after thyroid surgery with IONM technology has not been reported so far. This is the first research to observe the effects of IONM technology on early postoperative recovery quality of thyroid surgery patients undergoing by QoR-40 Questionnaire.

Our research found that the patients in the NEURO Group not only had a significant decrease in the global score of the QoR-40 Questionnaire on POD1 and POD3, but also had significantly lower scores in emotional state, physical comfort, physical independence and pain. When IONM applied on patients, the programs of anesthesia and surgery are different from traditional way. First, the patients in the NEURO Group only used 1*ED95 muscle relaxant during induction and no more muscle relaxant was allowed during whole procedure. This dose of muscle relaxant is less than half of that used in patients undergoing traditional thyroid surgery. As results, the clinical duration of muscle relaxant is shorter and recovery of neuromuscular block were faster in patients of NEURO group than those of CON group. Till now,

**Table 6. Comparison of postoperative adverse reactions on POD1.**

| | NEURO Group (N = 40) | CON Group (N = 40) | *P*-Value |
|---|---|---|---|
| **Nausea and vomiting** | 13 (32%)* | 4 (10%) | 0.029 |
| **Throat pain(VAS>3 point)** | 34 (85%) | 28 (70%) | 0.181 |
| **Hoarseness (n, %)** | 24 (60%) | 15 (38%) | 0.074 |
| **Drinking water cough** | 9 (22%) | 5 (12%) | 0.377 |
| **Wound pain(VAS>3 point)** | 33 (82%) | 29 (72%) | 0.422 |
| **Wound bleeding(need surgery)** | 0 (0%) | 0 (0%) | 1 |
| **Fatigue** | 20 (50%) | 18 (45%) | 0.823 |

Notes:

* *P*<0.05 (compared to the CON Group)

Data are presented as number (%)

there is no evidence about the postoperative recovery and the depth of neuromuscular block during operation in patients undergoing thyroid surgery. However, in laparoscopic surgery, Koo at all [18, 19] demonstrate that deep muscle relaxants can improve the postoperative satisfaction of patients. This hints that the lower dose of muscle relaxant may be one of the reasons why IONM decreases patients' QoR40 scores after surgery. Next, the endotracheal tube used in IONM has a wider outer diameter and balloon inflation size than that used in traditional way. In our study, outer diameters of tubes are 8.8mm and 10.5mm for female and male patients respectively in NEURO group. In control group, the outer diameters of tubes are 8.7mm and 9.5mm for female and male patients. The wider tube probably bring more nociceptive stimulation in the procedure of intubation and extubation. In addition, the balloon inflation diameter is also larger in NURO group than that in CON group. For female patients, the balloon inflation diameter is 23mm in NEURO group and 19mm in the CON Group. For male patients, the balloon inflation diameter is 26mm in NEURO group and 22mm in the CON Group. Finally, the surgeon will use the probe to detect the nerve activities by intermittently stimulating patients' vocal cords. If these stimulations deteriorate the patients' satisfaction after surgery, there is still no evidence.

Our results also show that more events of PONV in the NEURO Group, compared with the CON Group. The patients in the NEURO Group induced only 1*ED95 muscle relaxant during induction and no muscle relaxant were added during the operation. In order to avoid unexpected movement, more opioids or sedatives would be used in these patients. In our research, the dosage of remifentanil in the NEURO Group patients increased significantly, and the thyroid surgery time was shorter, the accumulation of these opioids in patients will eventually leads to adverse reactions such as nausea and vomiting after surgery. Nonetheless, whether these factors will increase the incidence of PONV needs further research. In our research, the total hospitalization cost of the patients in the NEURO Group was higher than those of the CON Group, and we believe that it was the additional costs incurred by the NIM endotracheal tube and never probe that greatly increased the total hospitalization costs of these patients.

IONM technology can effectively prevent the RLNP, which make this technology gradually accepted by surgeon. However, our study confirmed that IONM technology does lead to a decrease in the early recovery quality. Although many new technologies will provide us with many conveniences at the beginning of their advent, at the same time they will also bring many new problems. This research proposed some adverse consequences of IONM technology from the perspective of anesthesia, and suggested that we need to explore a set of anesthesia programs to improve the early recovery quality of thyroid surgery patients.

Although our research demonstrates that IONM technology reduced early recovery quality of thyroid surgery, it did not negate this technology, but to improve this technology. It is necessary to optimize anesthesia protocol to improve early recovery quality, and it also put forward directions for future research. In order to cooperate with surgeons to improve this technology, it is particularly important to explore a set of anesthesia programs to improve the early recovery quality. Our future research will focus on explore how to improve the early recovery quality of thyroid surgery patients with IONM technology without affecting the operation of surgeon.

There are some limitations in this research. First, we didn't measure the train-of-four ratio and balloon pressure in this study, so we haven't direct evidences to say that the lower QoR40 score is associated with the smaller dose of muscle relaxant and larger size of inflation balloon used in patients undergoing thyroid surgery with IONM. This study was not designed to identify the factors involved in quality of recovery and IONM used in surgery. Our objective is to observe the effects of IONM technology on recovery quality in patients after thyroid surgery. So, we believe our results are reliable for our conclusion. We need further clinical trials to

study why patients undergoing thyroid surgery with IONM suffer the worse postoperative recovery. Second, this research was not a blinded study. This is because the surgical method must be informed to the patients in detail before surgery, which cannot be concealed, and the NEURO Group also needs NIM endotracheal tube and never probe, so the anesthesiologist must also know it. Furthermore, our research did not distinguish between male and female genders. The recovery quality of male and female patients after thyroid surgery may be different. Although this does not affect the results of this study, future research should further compare the early recovery quality between different genders after the use of IONM technology. Lastly, our research concluded that the patients in the NEURO Group were less cost-effective. This is because the primary outcome of the study was the QoR-40 score, after sample evaluation according to this indicator, only 40 patients were included in each group, and all 80 patients did not suffer from RLNP eventually. With the increase of the risk of RLNP, the potential cost benefits of IONM technology will also increase. If the recurrent laryngeal nerve is injured due to the absence of IONM technology, the cost of NIM endotracheal tube and nerve probe is insignificant compared with the follow-up cost of treating RLNP and other complications.

## Conclusion

After thyroidectomy, the patients using IONM suffer worse quality of recovery, more risk of PONV and increased medical expense.

## Supporting information

**S1 Table. The quality of recovery-40 (QoR-40) questionnaire.**
(PDF)

**S1 File. Protocol for publication (Chinese version).**
(PDF)

**S2 File. Protocol for publication (English version).**
(PDF)

**S3 File. CONSORT 2010 checklist of information to include when reporting a randomised trial.**
(PDF)

## Author Contributions

**Conceptualization:** Haocong Chen.

**Data curation:** Haocong Chen.

**Formal analysis:** Haocong Chen.

**Funding acquisition:** Zhijun Lu.

**Investigation:** Haocong Chen.

**Methodology:** Haocong Chen.

**Project administration:** Haocong Chen.

**Resources:** Haocong Chen.

**Software:** Haocong Chen.

**Supervision:** Haocong Chen.

**Validation:** Haocong Chen.

**Visualization:** Haocong Chen.

**Writing – original draft:** Haocong Chen.

**Writing – review & editing:** Zhijun Lu.

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
