## [Decision Letter · Decision Letter 0]

11 Jul 2023

PONE-D-23-11841Effects of intraoperative neuromonitoring (IONM) technology on early recovery quality in patients after thyroid surgery: a randomized controlled trialPLOS ONE

Dear Dr. LU,

Thank you for submitting your manuscript to PLOS ONE. After careful consideration, we feel that it has merit but does not fully meet PLOS ONE’s publication criteria as it currently stands. Therefore, we invite you to submit a revised version of the manuscript that addresses the points raised during the review process.

We look forward to receiving your revised manuscript.

Kind regards,

Luigi La Via

Academic Editor

PLOS ONE

Journal Requirements:

6. We note that the original protocol that you have uploaded as a Supporting Information file contains an institutional logo. As this logo is likely copyrighted, we ask that you please remove it from this file and upload an updated version upon resubmission.

**Additional Editor Comments:**

- Please revise according to the indications given by the reviewers. In particular, please address all the issues underlined by reviewer 1.

Reviewers' comments:

Reviewer's Responses to Questions

**Comments to the Author**

1. Is the manuscript technically sound, and do the data support the conclusions?

Reviewer #1: No

Reviewer #2: Yes

2. Has the statistical analysis been performed appropriately and rigorously? 

Reviewer #1: Yes

Reviewer #2: Yes

3. Have the authors made all data underlying the findings in their manuscript fully available?

Reviewer #1: No

Reviewer #2: Yes

4. Is the manuscript presented in an intelligible fashion and written in standard English?

Reviewer #1: No

Reviewer #2: Yes

5. Review Comments to the Author

Reviewer #1: The authors conducted a randomized controlled study comparing the recovery of the patients after receiving a regular dose versus a low dose of rocuronium during the IONM in thyroid surgery. There are several concerns. First, it is unclear whether the authors wanted to investigate the effect of the low-dose rocuronium or the impact of IONM on the recovery.

Major

1. Figure 2 should be re-drawn to have the y-axis scale from 0. If it is drawn as it is now (e.g., QoR score begins from 150), the difference between the two groups looks exaggerated.

2. The results section is very painful to read. Lines 224 – 233 have too many numbers, and it isn't easy to understand the significance. Please focus on the positive findings with p values.

3. Patients in the two groups had different doses of rocuronium, and only the NEURO group underwent IONM. The difference in the results is from the difference in the dose of rocuronium or the use of IONM?

4. The authors suggest that the low dose of rocuronium is the cause of the low questionnaire scores in the IONM group. Despite the references the authors suggested, the assertion is not well backed by solid evidence. The reference paper by Koo was about laparoscopic abdominal surgery, which is quite different from thyroid surgery in terms that abdominal surgery requires deeper relaxation. Furthermore, the hemodynamics in the two groups in the current study are not different, suggesting a different dose of rocuronium was not critical to the participants.

5. The results of ToFr measurement should be demonstrated to assert that the dose of rocuronium made a difference in the questionnaire scores of the two groups.

6. The authors claim that the quality of life of the NEURO group is worse than the control group because the QoR score in the NEURO group is lower than the control group. Although the scores were statistically different, the difference in the absolute number was not so impressive (e.g., 160 vs 170 on POD1, 170 vs 174 on POD3). Does this difference is significant enough to conclude so?

7. Does the balloon size matter or balloon pressure matter? If the balloon pressure was controlled well, balloon size might not affect the tracheal symptoms. If the balloon size mattered, the dose of rocuronium is not important, and what mattered was just a tube.

Minor

1. English editing is mandatory.

2. Questionnaires should be given for the readers can understand the results better.

3. Detailed descriptions of the questionnaires should be moved from introduction to Methods.

4. Line 156-163 is unnecessary.

5. Line 335. "cost of the patients in the NEURO Group was less." It looks like a mistake because Table 5 shows that the cost in the NEURO group was higher.

6. Line 239-242 is too lengthy. The difference in MAP was already explained in Methods.

7. Line 267-287 is just a repetition of Introduction.

Reviewer #2: Thank you for the opportunity to read this manuscript on the effects of intraoperative neuromonitoring technology on early recovery quality in patients after thyroid surgery. The paper is interesting and sound. However, I have some comments to make:

- Line 75. Before introducing neuromonitoring, authors should report the increasing development of intraoperative monitoring in anesthesia with the use of artificial intelligence (doi: 10.1097/ALN.0000000000002960 - doi: 10.1016/j.bjane.2015.09.001 - doi: 10.3390/jcm11020392). Please briefly discuss and add these 3 references.

- Why did authors include only patient aged between 20 and 70 years? Please explain.

- Line 135. Who was aware of the group allocation?

- Line 174-175. Please replace "is" with "was".

- Line 212. Please replace decline with declined.

6. PLOS authors have the option to publish the peer review history of their article (what does this mean?). If published, this will include your full peer review and any attached files.

Reviewer #1: No

Reviewer #2: No

---

## [Author Response · Author response to Decision Letter 0]

19 Aug 2023

Dear editors and reviewers,

 Thank you very much for your comments and professional advice. These opinions help to improve academic rigor of our article. Based on your suggestion and request, we have made corrected modifications on the revised manuscript. Besides, our study’s original protocol and supporting information were uploaded to the editors when we submitting our revised manuscript. Furthermore, we would like to show the details as follows:

Reviewer 1#

Major

1.Figure 2 should be re-drawn to have the y-axis scale from 0. If it is drawn as it is now (e.g., QoR score begins from 150), the difference between the two groups looks exaggerated.

The author’s answer: We agree your opinion. All database have been shown in Table 2, so we think Figure 2 maybe a little bit repetition. So we deleted the Figure 2 .

2.The results section is very painful to read. Lines 224 – 233 have too many numbers, and it isn't easy to understand the significance. Please focus on the positive findings with p values.

The author’s answer: We agree that too many numbers isn't easy to understand the significance. Thank you for your reminder. We have deleted it.

3.Patients in the two groups had different doses of rocuronium, and only the NEURO group underwent IONM. The difference in the results is from the difference in the dose of rocuronium or the use of IONM?

The author’s answer: We have no evidence to conform that the lower dose of rocuronium leads to worse quality of recovery. This study is to compare the different qualities of recovery between thyroid surgery in traditional way and with IONM. Compared to traditional way, procedure with IONM have some different points, including A smaller dose of muscle relaxant, intubation of endotracheal tube with bigger outer diameter and intermittent stimulation on vocal cord. All of these are possible involved in decreased QoR40 scores in NRURO group. We have modified words expression in some paragraphs of introduction and discussion.

4.The authors suggest that the low dose of rocuronium is the cause of the low questionnaire scores in the IONM group. Despite the references the authors suggested, the assertion is not well backed by solid evidence. The reference paper by Koo was about laparoscopic abdominal surgery, which is quite different from thyroid surgery in terms that abdominal surgery requires deeper relaxation. Furthermore, the hemodynamics in the two groups in the current study are not different, suggesting a different dose of rocuronium was not critical to the participants.

The author’s answer: We agree with your opinion that, till now, there have been no solid evidence in relationship between neuromuscular block and quality of recovery in thyroid surgery. We didn’t monitored neuromuscular block for the whole surgical procedure. So we cannot provide direct evidence in this study. This is one of our limitation. In revision, we’ve written this limitation in the part of discussion. Our objective is not to study the quality of recovery and the depth of neuromuscular block. In NEURO group, the patients received no more than half dose of rocuronium, compared to those in control group. So, in theory, the clinical duration of rocuronium is shorter and recovery of neuromuscular block is faster in patients of NEURO group. According to the paper by Koo, we think this is a possible cause. Besides, the harder endotracheal tube enwrapped by electrode with wider outer diameter and intermittent stimulation on cocal cord during surgery also probably lead to a lower QoR40 score. These need further researches. We’ve revised several paragraphs in manuscript to express more correctly.

5.The results of ToFr measurement should be demonstrated to assert that the dose of rocuronium made a difference in the questionnaire scores of the two groups.

The author’s answer: We just measured the TOFR at the beginning and end of the anesthesia, so we can estimate the timing of intubation and extubation. This is one of our limitations. We’ve written this in the part of discussion in revision. Our objective is to observe the effects of thyroid surgery with IONM on patients’ postoperative recovery. This clinical trial is not designed to study the relationship between the depth of neuromuscular block and quality of recovery. So we didn’t measure TOFR for whole surgery. We’ve revised several paragraphs in manuscript to express more correctly.

6.The authors claim that the quality of life of the NEURO group is worse than the control group because the QoR score in the NEURO group is lower than the control group. Although the scores were statistically different, the difference in the absolute number was not so impressive (e.g., 160 vs 170 on POD1, 170 vs 174 on POD3). Does this difference is significant enough to conclude so?

The author’s answer: The differences of QoR40 scores between Neuro group and Control groupare minor on POD1 and POD2, but these differences are really significant in statistic. The QoR40 questionnaire has been demonstrated as a reliable muti-dimensional assessment tool, which can evaluate patient’s health status after surgery and anesthsia. Our conclusion is drawn according to the statistical difference in QoR40 scorces.

7.Does the balloon size matter or balloon pressure matter? If the balloon pressure was controlled well, balloon size might not affect the tracheal symptoms. If the balloon size mattered, the dose of rocuronium is not important, and what mattered was just a tube.

The author’s answer: We agree with your opinion the balloon pressure is very impertant. In this study, we didn’t measure the balloon pressure. This is one of limitations of this study. In revision, we’ve written this limitation in part of discussion. Not only balloon size but also the tube outer diameter of IONM tube is larger than traditional one. The endotracheal tube with larger outer diameter is another possible facor to bring nociceptive stimulation during intubation and extubation. We think these factors probably affect patient’s satisfaction after surgery and we need more studies to demonstrate. 

Minor

1.English editing is mandatory.

The author’s answer: We have edited English of this article.

2.Questionnaires should be given for the readers can understand the results better.

The author’s answer: We have put the QoR-40 Questionnaires as supporting information at the end of the manuscript.

3.Detailed descriptions of the questionnaires should be moved from introduction to Methods.

The author’s answer: We have moved detailed descriptions of the questionnairesfrom introduction to Methods.

4.Line 156-163 is unnecessary.

The author’s answer: We agree that this is unnecessary of the study. We have deleted it.

5.Line 335. "cost of the patients in the NEURO Group was less." It looks like a mistake because Table 5 shows that the cost in the NEURO group was higher.

The author’s answer: We were really sorry for our careless mistakes. Thank you for your reminder. We have replaced“less” to “higher”.

6.Line 239-242 is too lengthy. The difference in MAP was already explained in Methods.

The author’s answer:We agree that this is repetition of the explanation of MAP. Thank you for your reminder. We have deleted it.

7.Line 267-287 is just a repetition of Introduction.

The author’s answer: We agree that this is repetition of Introduction. Thank you for your reminder. We have deleted it.

Reviewer 2#

1.Line 75. Before introducing neuromonitoring, authors should report the increasing development of intraoperative monitoring in anesthesia with the use of artificial intelligence (doi: 10.1097/ALN.0000000000002960 - doi: 10.1016/j.bjane.2015.09.001 - doi: 10.3390/jcm11020392). Please briefly discuss and add these 3 references.

The author’s answer: We sincerely appreciate the valuable comments. We report the increasing development of intraoperative monitoring in anesthesia with the use of artificial intelligence and we also have added more references (doi: 10.1097/ALN.0000000000002960 - doi: 10.1016/j.bjane.2015.09.001 - doi: 10.3390/jcm11020392) before introducing neuromonitoring in the revised manuscript. Please see line 93-96.

2.Why did authors include only patient aged between 20 and 70 years? Please explain.

The author’s answer: Our research in the first research, so we choose the adults, and the elderly and minors may have organ function degeneration or immaturity. We will study different ages of people in the follow-up research.

3.Line 135. Who was aware of the group allocation?

The author’s answer: Surgeon and anesthesiologist aware of the group allocation. This research was not a blinded study. This is because the surgical method must be informed to the patients in detail before surgery, which cannot be concealed, and the NEURO Group also needs NIM endotracheal tube and never probe, so the anesthesiologist must also know it. 

4.Line 174-175. Please replace "is" with "was".

The author’s answer: We have changed “is” to “was”. Please see line 167-168

5.Line 212. Please replace decline with declined.

The author’s answer: We have changed “decline” to “declined”. Please see line 211

 Thank you very much for your attention and time. Look forward to hearing from you.

Yours sincerely,

Lu Zhijun

August 20, 2023

---

## [Decision Letter · Decision Letter 1]

11 Sep 2023

Effects of intraoperative neuromonitoring (IONM) technology on early recovery quality in patients after thyroid surgery: a randomized controlled trial

PONE-D-23-11841R1

Dear Dr. LU,

We’re pleased to inform you that your manuscript has been judged scientifically suitable for publication and will be formally accepted for publication once it meets all outstanding technical requirements.

Kind regards,

Luigi La Via

Academic Editor

PLOS ONE

Additional Editor Comments (optional):

Reviewers' comments:

Reviewer's Responses to Questions

**Comments to the Author**

1. If the authors have adequately addressed your comments raised in a previous round of review and you feel that this manuscript is now acceptable for publication, you may indicate that here to bypass the “Comments to the Author” section, enter your conflict of interest statement in the “Confidential to Editor” section, and submit your "Accept" recommendation.

Reviewer #1: All comments have been addressed

2. Is the manuscript technically sound, and do the data support the conclusions?

Reviewer #1: Yes

3. Has the statistical analysis been performed appropriately and rigorously? 

Reviewer #1: I Don't Know

4. Have the authors made all data underlying the findings in their manuscript fully available?

Reviewer #1: Yes

5. Is the manuscript presented in an intelligible fashion and written in standard English?

Reviewer #1: Yes

6. Review Comments to the Author

Reviewer #1: Authors replied all the questions and revised the manuscript properly. No further questions or comments.

7. PLOS authors have the option to publish the peer review history of their article (what does this mean?). If published, this will include your full peer review and any attached files.

Reviewer #1: No

---

## [Editor Report · Acceptance letter]

18 Sep 2023

PONE-D-23-11841R1 

Effects of intraoperative neuromonitoring (IONM) technology on early recovery quality in patients after thyroid surgery: a randomized controlled trial 

Dear Dr. Lu:

I'm pleased to inform you that your manuscript has been deemed suitable for publication in PLOS ONE. Congratulations! Your manuscript is now with our production department. 

Kind regards, 

on behalf of

Dr. Luigi La Via 

Academic Editor

PLOS ONE